# ATP and large signaling metabolites flux through caspase-activated Pannexin 1 channels

Adishesh K Narahari[1]*, Alex JB Kreutzberger[2†‡], Pablo S Gaete[3†], Yu-Hsin Chiu[1§], Susan A Leonhardt[2], Christopher B Medina[4#], Xueyao Jin[2], Patrycja W Oleniacz[1], Volker Kiessling[2], Paula Q Barrett[1], Kodi S Ravichandran[4], Mark Yeager[2], Jorge E Contreras[3], Lukas K Tamm[2], Douglas A Bayliss[1]*

[1]Department of Pharmacology, University of Virginia, Charlottesville, United States; [2]Department of Molecular Physiology and Biological Physics, University of Virginia, Charlottesville, United States; [3]Department of Pharmacology, Physiology, and Neuroscience, Rutgers New Jersey Medical School, Newark, United States; [4]Department of Microbiology, Immunology, and Cancer Biology, University of Virginia, Charlottesville, United States

*For correspondence:
akn4uq@virginia.edu (AKN);
bayliss@virginia.edu (DAB)

[†]These authors contributed equally to this work

Present address: [‡]Department of Cell Biology, Harvard Medical School and Program in Cellular and Molecular Medicine, Boston Children's Hospital, Boston, United States; [§]Institute of Biotechnology and Department of Medical Science, National Tsing Hua University, Hsinchu, Taiwan; [#]Emory Vaccine Center and Department of Microbiology and Immunology, Emory University, Atlanta, United States

Competing interests: The authors declare that no competing interests exist.

**Abstract** Pannexin 1 (Panx1) is a membrane channel implicated in numerous physiological and pathophysiological processes via its ability to support release of ATP and other cellular metabolites for local intercellular signaling. However, to date, there has been no direct demonstration of large molecule permeation via the Panx1 channel itself, and thus the permselectivity of Panx1 for different molecules remains unknown. To address this, we expressed, purified, and reconstituted Panx1 into proteoliposomes and demonstrated that channel activation by caspase cleavage yields a dye-permeable pore that favors flux of anionic, large-molecule permeants (up to ~1 kDa). Large cationic molecules can also permeate the channel, albeit at a much lower rate. We further show that Panx1 channels provide a molecular pathway for flux of ATP and other anionic (glutamate) and cationic signaling metabolites (spermidine). These results verify large molecule permeation directly through caspase-activated Panx1 channels that can support their many physiological roles.

## Introduction

Pannexin 1 (Panx1) is a widely expressed homo-heptameric membrane channel that plays a critical role in numerous physiological and pathophysiological processes. Among others, this includes cell clearance after apoptosis (*Chekeni et al., 2010*; *Medina et al., 2020*; *Poon et al., 2014*), blood pressure regulation (*Billaud et al., 2011*; *Good et al., 2018a*), stroke (*Bargiotas et al., 2011*; *Good et al., 2018b*; *Thompson, 2015*), and neuropathic pain (*Bravo et al., 2014*; *Weaver et al., 2017*; *Zhang et al., 2015*). Individual Panx1 subunits consist of a four-transmembrane α-helical bundle with a cytoplasmic loop between TM2 and TM3; the N- and C-termini also reside on the cytoplasmic surface (*Baranova et al., 2004*; *Penuela et al., 2013*). These channels are broadly similar to a subset of mammalian large-pore ion channels that include connexin gap junctions (*Panchin et al., 2000*), calcium homeostasis modulator 1 (CALHM1) (*Siebert et al., 2013*; *Syrjanen et al., 2020*), and SWELL1 (LRRC8) (*Abascal and Zardoya, 2012*; *Deneka et al., 2018*). Despite a conserved subunit topology, recent cryoEM structures revealed that these different channels exist in a variety of oligomeric states (from hexameric to undecameric) (*Deng et al., 2020*; *Michalski et al., 2020*). Of interest here, activation of these channels has been associated with both ionic current and large molecule permeation. In particular, channel activation is associated with release of metabolites (often

ATP) that are critical for their roles in intercellular signaling (*Bao et al., 2004*; *Chekeni et al., 2010*; *Lutter et al., 2017*; *Medina et al., 2020*; *Siebert et al., 2013*; *Taruno, 2018*).

In the two decades since their discovery, much has been learned about the functional properties of Panx1 channels. Panx1 ionic currents have occasionally been detected under unstimulated conditions (*Ma et al., 2012*; *Romanov et al., 2012*; *Ruan et al., 2020*), but that basal activity does not appear to be associated with large molecule permeation (*Romanov et al., 2012*). By contrast, transmembrane permeation of large fluorescent dyes and ATP occurs when Panx1 is activated by various mechanisms (e.g., stretch, elevated external K⁺, ionotropic and metabotropic receptor signaling, and caspase-mediated cleavage at a C-terminal site) (*Beckel et al., 2015*; *Chekeni et al., 2010*; *Chiu et al., 2017*; *Iglesias et al., 2008*; *Silverman et al., 2009*; *Weilinger et al., 2016*). Among these, caspase-mediated activation is a well-characterized mechanism in which cleavage of the channel C-tails, as occurs during apoptosis or pyroptosis, is accompanied by uptake of dyes indicative of cell death (e.g., Yo-Pro-1 and To-Pro-3) and release of ATP and UTP that serve as 'find-me' signals to direct phagocytic clearance of cell corpses (*Chekeni et al., 2010*; *Chiu et al., 2017*; *Poon et al., 2014*; *Qu et al., 2011*; *Yang et al., 2015*). Moreover, Panx1 cleavage elicits efflux of additional metabolites from apoptotic cells, which function as local 'good-bye' signals with anti-inflammatory, wound healing, and cell proliferative actions (*Medina et al., 2020*). Notably, all studies to date have been performed using intact cells. Therefore, this previous work does not demonstrate that large molecule permeation occurs directly via the channel itself or exclude a secondary release mechanism. Moreover, recently available Panx1 channel structures from both intact (inactive) and caspase-cleaved (activated) channels reveal a central pore with an extracellular constriction that has a diameter of ~9 Å, which would appear to be too small to enable large molecule permeation (*Deng et al., 2020*; *Jin et al., 2020*; *Michalski et al., 2020*; *Mou et al., 2020*; *Qu et al., 2020*; *Ruan et al., 2020*). Thus, the ability of caspase-activated Panx1 to support metabolite release remains uncertain.

In this study, we developed a proteoliposome system for reconstitution of purified Panx1 and channel activation by caspase-mediated cleavage. By combining lipid bilayer electrophysiology, flow cytometry, total internal reflection fluorescence (TIRF) microscopy, and radioactive metabolite uptake assays, we characterized the large molecule permeation properties of cleavage-activated Panx1. We determined that the caspase-activated Panx1 channel forms a permeation pathway that favors anionic over cationic molecules and supports flux of ATP and other important signaling metabolites, such as glutamate and spermidine.

## Results

For these studies, we used the Panx1 ortholog from *Xenopus tropicalis* (i.e., frog Panx1 and fPanx1) that has been examined in recent structural studies (*Deng et al., 2020*; *Michalski et al., 2020*).

### Preparation of Panx1 proteoliposomes

An fPanx1-enhanced green fluorescent protein (eGFP) fusion construct, containing a thrombin cleavage site and a Strep-tag, was inserted into pFastBac for expression in Sf9 cells (*Figure 1A*) and purified by affinity chromatography and size-exclusion chromatography (SEC) (*Figure 1—figure supplement 1A,B*). Purified fPanx1-eGFP was incorporated into proteoliposomes that appeared in fractions 1–3 of a co-floatation assay on a Nycodenz density gradient (*Figure 1B,C*; *Hernandez et al., 2012*). When analyzed by negative-stain electron microscopy, we observed a normal distribution of circular proteoliposomes in projection images, with the expected mean diameter of 97.3 ± 22 nm (*Figure 1D*, *Figure 1—figure supplement 1C*). Panx1 cleavage by caspases is a prominent mechanism for channel activation. Therefore, we verified that purified fPanx1 incorporated into proteoliposomes was cleaved at the expected sites following overnight incubation with recombinant Caspase-3 (Casp3) (*Figure 1E*). Note that although the channel can be cleaved at multiple sites, cleavage at the C-terminal site is necessary and sufficient for channel activation (*Chekeni et al., 2010*; *Chiu et al., 2017*; *Sandilos et al., 2012*).

### Properties of caspase-activated fPanx1 in cells and lipid bilayers

In recordings from transfected HEK293T cells, we found that fPanx1 was basally silent and could be activated by intracellular dialysis with Casp3 to generate whole cell currents with an outwardly-rectifying current–voltage profile (*Figure 1—figure supplement 2A,B*) similar to that observed with

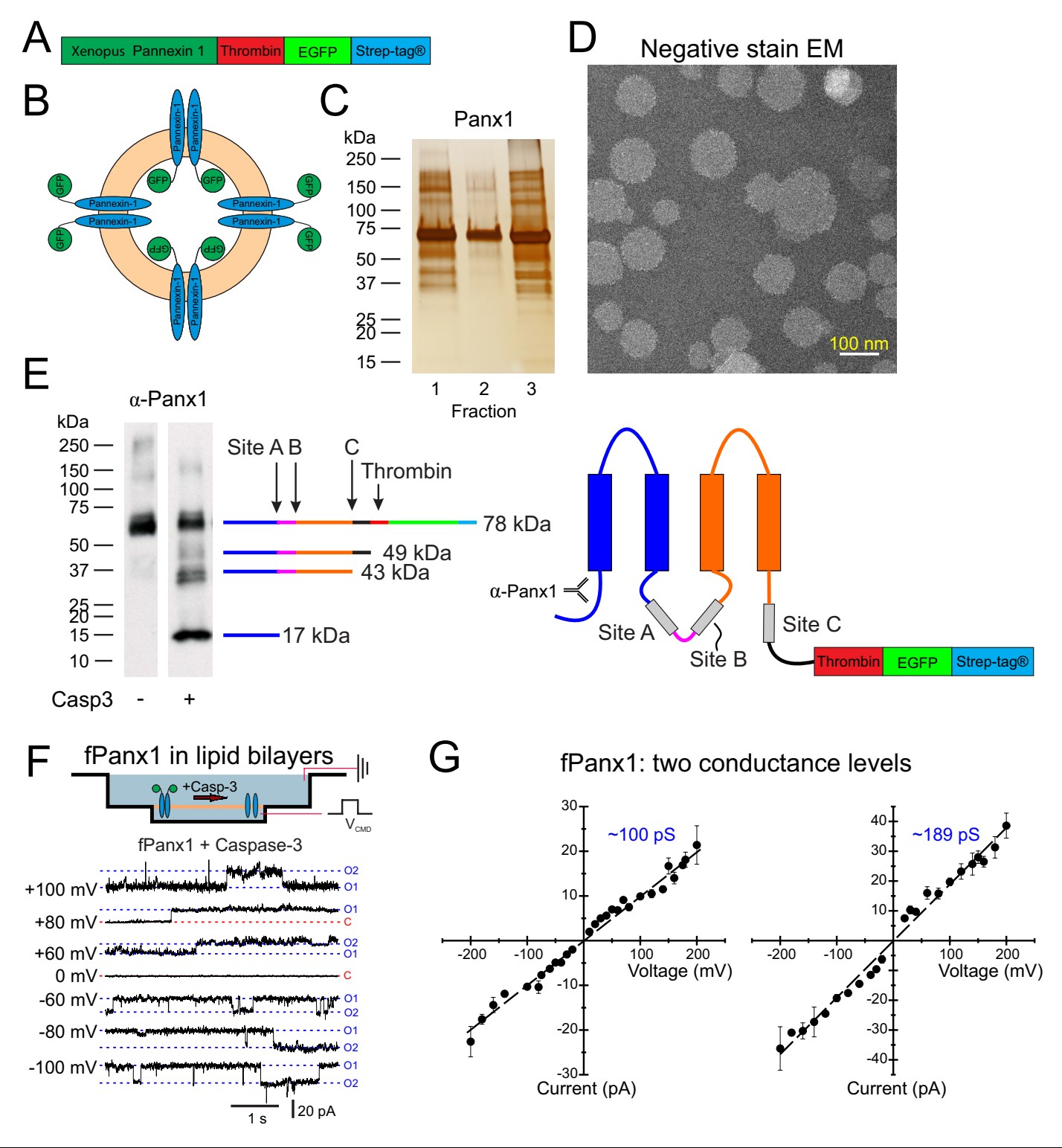

**Figure 1.** Liposome reconstitution and caspase cleavage-activation of purified *Xenopus* Pannexin 1 (fPanx1). (A) Schematic of recombinant fPanx1 construct incorporating a thrombin cleavage site, (eGFP), and a Strep-Tag. (B) Schematic of proteoliposomes containing fPanx1-eGFP fusion proteins, in either orientation. (C) fPanx1-containing fractions from Nycodenz co-floatation assay (fractions 1–3, 20 µL each) were run on a polyacrylamide gel and analyzed by silver stain. (D) Negative stain electron microscopy image of fPanx1-eGFP proteoliposomes extruded at 100 nm shown at 29,000× magnification. (E) Western blot of fPanx1-eGFP from proteoliposomes after overnight incubation in the absence and presence of recombinant Caspase-3 (Casp3). The schematics illustrate the location of: caspase cleavage sites, N-terminally directed α-Panx1 antibody, and corresponding cleavage

*Figure 1 continued on next page*

*Figure 1 continued*

products. Note that residual thrombin from activating recombinant caspase cleaves at its cognate C-terminal site. (F) *Upper*: Schematic of fPanx1-eGFP embedded in bilayer in recording chambers in NanIon Orbit mini; only channels in the orientation shown are activated by recombinant Casp3 added to the chamber. Positions of recording and ground electrodes are depicted. *Lower*: Recordings of purified fPanx1 channels in planar lipid bilayers at the indicated voltages following activation by recombinant Casp3; current levels are indicated that correspond to closed (**C**) state and apparent openings of one or two channels (O1, O2). (**G**) Unitary current voltage relationships for caspase-activated fPanx1 show two different conductance states (N = 3 bilayers with each conductance level). Numerical data for conductance measurements from lipid bilayer recordings are presented in *Figure 1—source data 1*.

The online version of this article includes the following source data and figure supplement(s) for figure 1:

**Source data 1.** Conductance measurements of *Xenopus* Pannexin 1 (fPanx1) in lipid bilayers.
**Figure supplement 1.** Purification of *Xenopus* Pannexin 1 (fPanx1).
**Figure supplement 1—source data 1.** Diameters of *Xenopus* Pannexin 1 (fPanx1) proteoliposomes.
**Figure supplement 2.** In vitro electrophysiology of *Xenopus* Pannexin 1 (fPanx1) in mammalian cells.
**Figure supplement 2—source data 1.** In vitro electrophysiology of *Xenopus* Pannexin 1 (fPanx1).
**Figure supplement 2—source data 2.** *Xenopus* Pannexin 1 (fPanx1) unitary conductance.
**Figure supplement 3.** Caspase has no effect on lipid bilayers that do not contain Xenopus Pannexin 1 (fPanx1).

cleavage-activated human Panx1 (hPANX1) channels (*Figure 1—figure supplement 2C*; *Chiu et al., 2017*; *Sandilos et al., 2012*). These currents were blocked by the Panx1 inhibitor carbenoxolone (CBX). In addition, and also similar to hPANX1, single fPanx1 channels were silent in excised inside-out patches until activated by Casp3 (*Chiu et al., 2017*); the cleavage-activated channels were inhibited by CBX and displayed a unitary conductance of 91.4 ± 13.5 pS (N = 3, *Figure 1—figure supplement 2D,E*). Thus, whole cell and single channel properties of fPanx1 are essentially identical to hPANX1 after caspase activation (*Chiu et al., 2017*).

We next used lipid bilayer recordings to characterize single channel properties of purified fPanx1 after caspase cleavage. The purified channels were added to DPhPC (1,2-diphytanoyl-sn-glycero-3-phosphocholine, 4ME16:0 PC) lipid bilayers and Casp3 was added to the bath to activate those channels with their C-terminus facing the *cis* side of the chamber (*Figure 1F*). Prior to application of caspase, large 'flickery' currents were occasionally observed at extreme potentials (e.g., +140 mV; *Figure 1—figure supplement 3A*), and this often signified successful incorporation of fPanx1 into the bilayer. However, before caspase addition, fPanx1 channel activity was not observed at physiological potentials, even in a bath solution containing 200 mM $K^+$ (*Figure 1—figure supplement 3B*). After caspase addition, channel currents with discrete openings and closings became apparent across a wide range of physiological voltages (*Figure 1F*, *Figure 1—figure supplement 3B*); this activity was not seen when caspase was applied to plain lipid bilayers (i.e., with no channel added; *Figure 1—figure supplement 3C*), indicating that measured currents reflected cleavage activation of the bilayer-incorporated fPanx1 channels.

The properties of caspase-activated fPanx1 channels in bilayers were similar, but not identical, to those of heterologously expressed recombinant fPanx1 channels recorded in mammalian cells (*Figure 1—figure supplement 2D,E*). The individual bilayers usually included multiple active channels that presented with either of two distinct conductance levels: one set had a unitary conductance of ~100 pS, similar to that recorded in mammalian cells, and the second displayed a larger unitary conductance of ~189 pS (*Figure 1G*). For both, the open channel I-V relationships were approximately ohmic (*Figure 1G*), unlike the outwardly-rectifying single channel conductance observed for recombinant caspase-activated Panx1 in mammalian cells (*Chiu et al., 2017*). These differences may be due to species variants and/or the fact that the bilayer recordings were obtained in symmetrical solutions across a non-native membrane composed of non-physiological lipids (e.g., without cholesterol, phosphatidylinositol 4,5-bisphosphate ($PIP_2$), etc.). Nevertheless, the purified fPanx1 channel is clearly activated by caspase.

## Panx1 forms a dye permeable pore

Caspase-mediated activation of Panx1 is associated with transmembrane flux of various metabolites and fluorescent dyes (*Chekeni et al., 2010*; *Medina et al., 2020*; *Nielsen et al., 2020*; *Qu et al., 2011*). Importantly, all previous experiments have been performed with Panx1 expressed in cells, either endogenously or heterologously. Therefore, a secondary mechanism of Panx1-dependent

large molecule flux could not be excluded. To address this directly, we tested whether large molecule permeation can occur upon caspase cleavage-based activation of fPanx1 reconstituted in proteoliposomes (containing phosphatidylcholine, total brain lipid extract, cholesterol, and PIP$_2$).

In a first set of studies, we examined whether caspase-cleaved fPanx1 is capable of forming a dye permeable pore. After overnight treatment with Casp3 (or buffer alone), liposomes were incubated with the anionic fluorescent dye, Sulforhodamine B (SR-B), and eluted through G25 spin columns for analysis on a fluorescence plate reader. Uptake of SR-B into liposomes was observed only when they contained fPanx1 and were treated with caspase. In addition, dye uptake was inhibited when caspase-treated fPanx1-containing proteoliposomes were treated with CBX before and during the dye incubation period (*Figure 2—figure supplement 1*).

To quantify this dye uptake, and verify that the dye was indeed associated with proteoliposomes containing fPanx1-GFP, we analyzed SR-B uptake in Casp3-treated and control proteoliposomes by ImageStream flow cytometry (*Figure 2A*). As depicted in the representative images (*Figure 2B*), SR-B was only observed in proteoliposomes that were treated with Casp3. A high GFP signal was present in the proteoliposomes that were not exposed to caspase, verifying the presence of GFP-tagged fPanx1 in those proteoliposomes that did not accumulate SR-B. Normalized frequency histograms of the mean fluorescence intensity (MFI) for GFP and SR-B quantify results from this exemplar experiment. Compared to untreated proteoliposomes, a larger fraction of Casp3-treated proteoliposomes contained SR-B (83.6% vs. 20.4%; *Figure 2C*) and at ~10-fold higher MFI levels. We found a concomitant, albeit modest, leftward shift in the GFP signal after caspase treatment, reflecting cleavage of the outward-facing fPanx1 C-tail (exposed to caspase) and the associated removal of the C-terminal GFP tag (*Figure 2C*); the retained GFP signal in the caspase-treated proteoliposomes may reflect uncut channels or channels with their GFP-tagged C-tails oriented into the proteoliposome. In multiple independent experiments and proteoliposome preparations, Casp3 treatment of fPanx1 proteoliposomes resulted in a decrease in GFP signal and a concomitant increase in SR-B signal (*Figure 2D*, *Figure 2—figure supplement 2*), indicating that caspase-cleaved fPanx1 channels generate a pore that is large enough for permeation of the SR-B dye.

## Panx1 is an anion-preferring molecular sieve

We exploited the dye permeation properties of fPanx1 channels to develop a single-particle assay, in which TIRF microscopy was used to characterize the kinetics of dye flux and the charge/size determinants of fPanx1 permeants in proteoliposomes (*Figure 3A*; *Farsi et al., 2016*).

For these experiments, liposomes were pre-filled with SR-B and visualized by TIRF microscopy before and after the addition of Casp3 to the bath solution. For each liposome, we recorded changes in the fluorescence intensity of both GFP, representing fPanx1 C-tail channel cleavage, and SR-B, representing dye release from the proteoliposome (*Figure 3B*). In a prominent subset of fPanx1-containing proteoliposomes, both GFP and SR-B fluorescence decreased following Casp3 application (*Figure 3B,C*, *purple*). For other fPanx1-containing proteoliposomes, the GFP fluorescence was not reduced by Casp3, suggesting that the GFP-tagged fPanx1 C-tail was oriented into the proteoliposome and remained uncleaved; in those cases, the SR-B fluorescence was also unchanged (*Figure 3C*, *green*). Likewise, SR-B fluorescence was unaffected following Casp3 application in particles that were devoid of GFP fluorescence, likely representing liposomes that did not incorporate any fPanx1-GFP (*Figure 3C*, *red*). In other controls, SR-B fluorescence was retained in liposomes, either with or without GFP fluorescence (i.e., with or without fPanx1) when they were not exposed to Casp3; and SR-B fluorescence was also unaffected by Casp3 in liposomes that did not contain fPanx1 (i.e., empty liposomes) (*Figure 3C*). Thus, this TIRF-based assay verifies that caspase cleavage activation of fPanx1 elicits dye efflux from fPanx1-containing proteoliposomes.

The TIRF assay was also used to examine the kinetics of dye release from proteoliposomes, thereby enabling flux measurements between fluorescent dyes of different charge and size. To examine charge effects, we assessed the release of a cationic dye, Rhodamine B (RhB), which is slightly smaller than the anionic SR-B (480 Da vs. 559 Da; *Figure 3—figure supplement 1*). Like SR-B, RhB was released from proteoliposomes, and this release was also dependent on caspase-mediated cleavage of fPanx1 (*Figure 3D,E*). The rate of dye efflux, obtained from fits to mono-exponential decay curves of the fluorescent signal for each proteoliposome, was positively correlated with the fraction of fPanx1 cleavage (i.e., reduction in GFP fluorescence) for both SR-B and RhB dyes

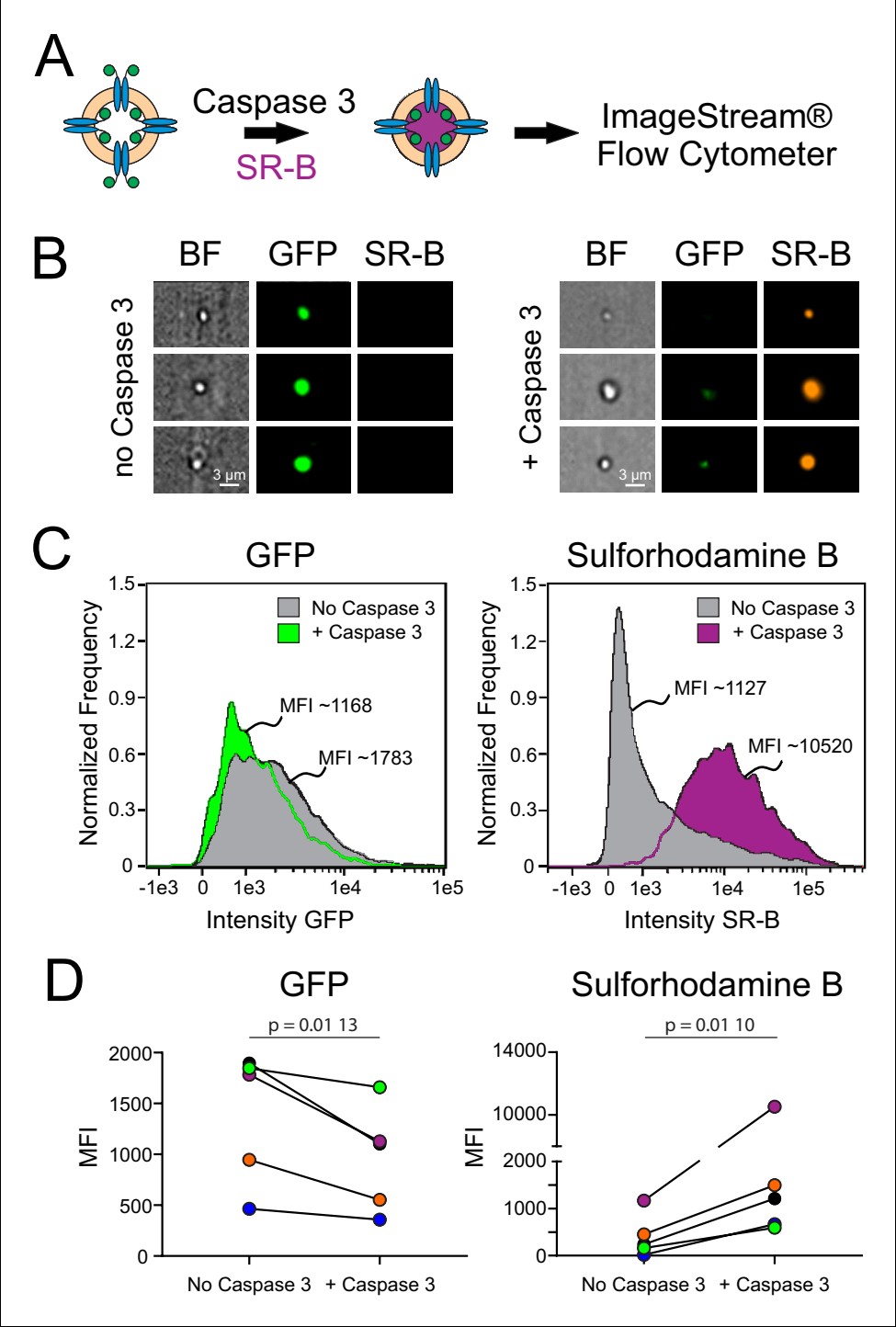

**Figure 2.** Dye uptake in caspase-treated *Xenopus* Pannexin 1 (fPanx1)-containing proteoliposomes. (**A**) Proteoliposomes containing fPanx1 were incubated overnight with recombinant Caspase-3 (Casp3) and for 3 hr with sulforhodamine B (SR-B) dye prior to analysis by ImageStream flow cytometry. (**B**) Representative images of liposomes treated with vehicle (left) or Casp3 (right) viewed by brightfield, or on channels for (GFP) and SR-B fluorescence. (**C**) Frequency distributions of fluorescence intensity show that proteoliposomes treated with recombinant Casp3 show a reduction in mean GFP intensity (*left*) and an increase in SR-B intensity (*right*), relative to vehicle treated proteoliposomes. (**D**) Mean fluorescence intensity (MFI) of GFP fluorescence (*left*) (p=0.0096) and SR-B fluorescence (*right*) (p=0.0104) before and after caspase treatment are shown with individual experiments

*Figure 2 continued on next page*

*Figure 2 continued*

depicted according to the color shown (N = 5). A paired t-test was performed. Numerical data for changes in GFP and SR-B MFI are presented in *Figure 2—source data 1*.

The online version of this article includes the following source data and figure supplement(s) for figure 2:

**Source data 1.** Mean fluorescence intensity (MFI) of *Xenopus* Pannexin 1 (fPanx1) proteoliposomes before and after Caspase-3 treatment.

**Figure supplement 1.** Bulk dye uptake in caspase-treated *Xenopus* Pannexin 1 (fPanx1)-containing proteoliposomes.

**Figure supplement 1—source data 1.** Bulk dye uptake in caspase-treated *Xenopus* Pannexin 1 (fPanx1)-containing proteoliposomes.

**Figure supplement 2.** (GFP) and sulforhodamine-B mean fluorescence intensity (MFI) in caspase-treated *Xenopus* Pannexin 1 (fPanx1)-containing proteoliposomes.

**Figure supplement 2—source data 1.** (GFP) and sulforhodamine-B mean fluorescence intensity (MFI) in caspase-treated *Xenopus* Pannexin 1 (fPanx1)-containing proteoliposomes.

**Figure supplement 2—source data 2.** Flow cytometry dot plots of ImageStream experiments.

---

(*Figure 3F*). Notably, despite the smaller size of RhB, the efflux rate was significantly slower than for SR-B due to the cationic charge (*Figure 3F*).

To examine the effect of size on efflux rates, we tested two additional anionic fluorescent dyes of increasing size, Alexa 594 and Alexa 555 (880 Da and 980 Da, respectively; *Figure 3G*, *Figure 3—figure supplement 1*). These larger dyes permeated caspase-cleaved fPanx1 at rates substantially slower than the smaller SR-B (*Figure 3H*). Of note, however, despite being approximately two times larger, these anionic dyes transited the channel at rates comparable to the cationic dye RhB (*Figure 3H*). Finally, we also examined two larger dyes (ATTO 550 (+), 1363 Da; and Dextran 3000 (−), 3000 Da; *Figure 3—figure supplement 1*), and neither of those were able to permeate caspase-activated fPanx1 (*Figure 3G*). The permeation properties of these dyes were not related to the relative amounts of GFP cleavage, which were comparable in all cases (*Figure 3I*). Together, these data indicate that the channel favors anionic over cationic permeants and reveal that the pore size is sufficient to accommodate molecules up to 980 Da.

## Panx1 is a conduit for anionic and cationic metabolites

Panx1 is commonly characterized as an ATP-release channel (*Chekeni et al., 2010*; *Chiu et al., 2017*; *Chiu et al., 2018*; *Dahl, 2015*; *Medina et al., 2020*; *Taruno, 2018*) and recent work provides compelling evidence that caspase cleavage-based activation of Panx1 can also lead to release of multiple additional metabolites (e.g., spermidine) that mediate important intercellular signaling processes (*Medina et al., 2020*). To test whether ATP and other metabolites permeate directly through caspase-activated fPanx1 channels, we developed a filtration-based assay for uptake of radiolabeled metabolites in proteoliposomes (*Figure 4A*). We performed this assay with select metabolites that have been proposed to permeate cleavage-activated fPanx1 (*Medina et al., 2020*), including anionic ($\alpha$-[$^{32}$P]ATP and [$^3$H]-glutamate) and cationic ([$^3$H]-spermidine) metabolites (*Figure 4—figure supplement 1*). Indeed, we found that uptake of $\alpha$-[$^{32}$P]ATP, [$^3$H]-glutamate, and [$^3$H]-spermidine into proteoliposomes required caspase-activated fPanx1; uptake was not observed when liposomes did not contain fPanx1 or when the channel was not activated by Casp3 (*Figure 4B–D*). We also tested whether ATP and spermidine could flux via caspase-activated Panx1 when included together in the same assay, rather than independently. Under these conditions, we again found uptake of both ATP and spermidine into caspase-cleaved Panx1-containing proteoliposomes, albeit at slightly lower levels (*Figure 4—figure supplement 2*). These data indicate that cleavage-activated fPanx1 itself is sufficient to form a membrane conduit capable of conducting ATP and other important signaling metabolites.

## Discussion

Panx1 membrane channels are best known for their purported ability to support transmembrane flux of large molecules such as fluorescent dyes and, of more physiological relevance, metabolites and intercellular signaling molecules (e.g., most notably, ATP) (*Chekeni et al., 2010*; *Chiu et al., 2017*;

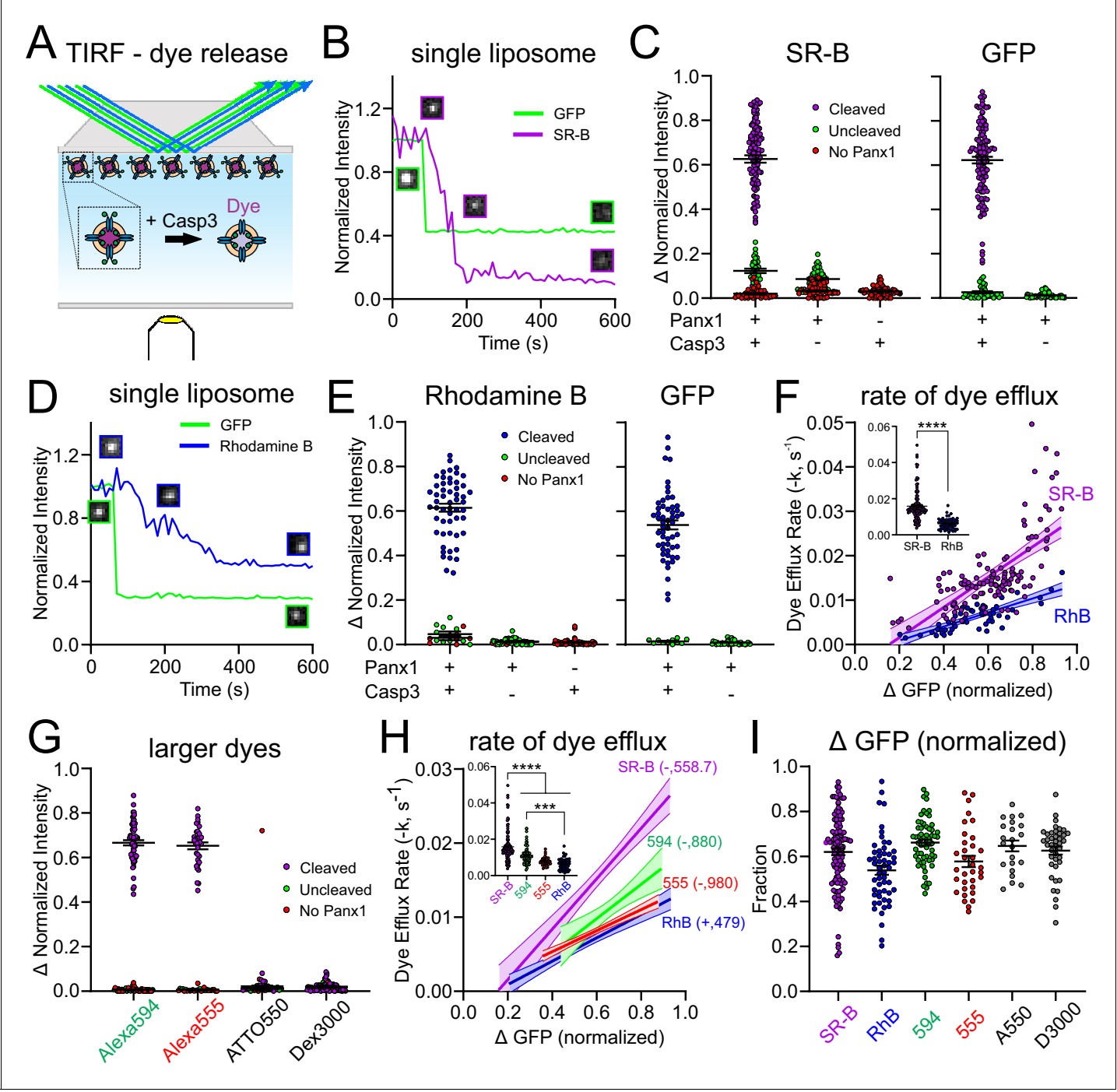

**Figure 3.** *Xenopus* Pannexin 1 (fPanx1) favors permeation of anionic dyes. (**A**) Schematic of experimental design to assay dye release kinetics from caspase-treated fPanx1-containing proteoliposomes by total internal reflection fluorescence (TIRF) microscopy. (**B**) Example fluorescence intensity traces for sulforhodamine B (SR-B, anionic dye, 559 Da) and (GFP) (caspase cleavage) over time in proteoliposomes after caspase treatment. Sample images of the proteoliposome fluorescence at the different time points are also provided. (**C**) Steady-state change in normalized fluorescence intensity for SR-B (*left*) and GFP (*right*) from fPanx1-GFP-containing proteoliposomes treated with either Caspase-3 (Casp3) or vehicle, or from empty liposomes (no fPanx1-GFP) treated with Casp3. Data from fPanx1-containing proteoliposomes were grouped according to whether they showed a reduction in GFP fluorescence (cleaved) or no change in GFP fluorescence (<10%, uncleaved) (N = 5 (+)Panx 1 (+)Casp3, N = 5 (+)Panx1 (−)Casp3, N = 4 (−)Panx1 (+) Casp3). (**D, E**) Fluorescence intensity traces (**D**) and steady-state change in normalized fluorescence intensity (**E**) for Rhodamine B (RhB, cationic dye, 479 Da) and GFP, as described for (**B, C**) (N = 5 (+)Panx1 (+)Casp3, N = 3 (+)Panx1 (−)Casp3, N = 4 (−)Panx1 (+)Casp3). (**F**) Efflux rates for SR-B and RhB were determined from fits of mono-exponential to the fluorescence intensity decay curves for individual caspase-treated proteoliposomes and plotted relative to the change in GFP fluorescence (i.e., fPanx1 cleavage); overlaid regression lines are depicted (with 95% confidence interval; slopes were

*Figure 3 continued on next page*

*Figure 3 continued*

significantly different, p=0.0014). Inset shows dye efflux rates for individual liposomes. Individual liposome dye efflux rates were analyzed by a Mann–Whitney test (p<0.0001) (G) Steady-state change in normalized fluorescence intensity for Alexa594 (anionic, 880 Da), Alexa555 (anionic, 980 Da), ATTO550 (cationic, 1363 Da), and Dextran 3000 (anionic, 3000 Da) from fPanx-GFP-containing proteoliposomes treated with Casp3 (Alexa 594 N = 6, Alexa 555 N = 5, ATTO 550 N = 3, Dextran 3000 N = 5). (H) Efflux rates for the indicated dyes represented by associated regression lines (with 95% confidence interval), with pairwise comparison of slopes: SR-B vs. RhB (p=0.0014), SR-B vs. 594 (p=0.16), SR-B vs. 555 (p=0.0074), 594 vs. 555 (p=0.15), 594 vs. RhB (p=0.16), 555 vs. RhB (p=0.53). Inset shows dye efflux rates for individual liposomes (ANOVA: $F_{3,267}$ = 36.18, p<0.0001; with Tukey's multiple comparisons test: ****p<0.0001; ***p=0.0002). (I) Relative change (upper) and rate (lower) of GFP fluorescence in cleaved fPanx1-containing proteoliposomes filled with the indicated dyes. Numerical data for all dye release data and GFP cleavage are provided in *Figure 3—source data 1*. The online version of this article includes the following source data and figure supplement(s) for figure 3:

**Source data 1.** Total internal reflection fluorescence (TIRF) imaging of dye release from proteoliposomes.
**Figure supplement 1.** Chemical structures of fluorescent dyes.

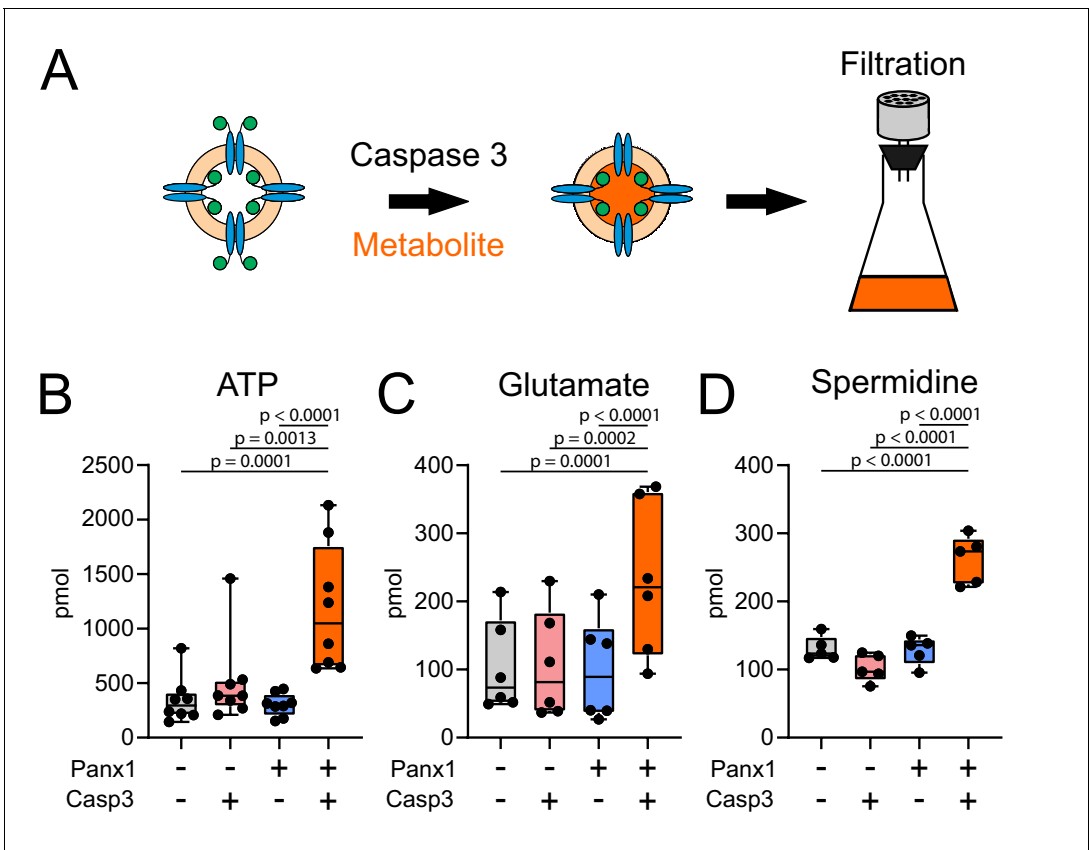

**Figure 4.** Caspase-cleaved *Xenopus* Pannexin 1 (fPanx1) is a conduit for metabolite release. (A) Schematic depicting experimental design for treating fPanx1-containing proteoliposomes with recombinant Caspase-3 (Casp3) overnight at 4°C before incubation with 4 µCi each of α[$^{32}$P]-ATP (B, 1 mM;~1:150000, hot:cold), [$^3$H]-Glutamate (C, 0.8 mM;~1:2000), and [$^3$H]-Spermidine (D, 8 µM;~1:24) for 3 hr and filtration using a Whatman GF/B filter. (B–D) Metabolites taken up by proteoliposomes under the indicated conditions for α[$^{32}$P]-ATP (N = 8), [$^3$H]-Glutamate (N = 6), and [$^3$H]-Spermidine (N = 5); molar quantities should not be compared between compounds due to different assay conditions. A box plot with the box depicting the quartiles/median, and lines drawn to points outside 25th/75th percentiles are shown. By repeated-measures one-way ANOVA (ATP: $F_{3,28}$ = 14.18, p<0.0001; Glutamate: $F_{3,20}$ = 18.29, p<0.0001; Spermidine: $F_{3,16}$ = 40.50, p<0.0001), with p-values provided from Tukey's multiple comparisons tests. Numerical data for individual metabolite flux are shown in *Figure 4—source data 1*. The online version of this article includes the following source data and figure supplement(s) for figure 4:

**Source data 1.** Metabolite flux in proteoliposomes.
**Figure supplement 1.** Chemical structures of metabolites.
**Figure supplement 2.** ATP and Spermidine can flux through Pannexin 1 concurrently.
**Figure supplement 2—source data 1.** ATP and spermidine concurrent flux in proteoliposomes.

*Yang et al., 2015*). The widespread acceptance of this idea is based on a wealth of data obtained from more intact systems, including those that implicated caspase-activated Panx1 as the likely transmembrane flux pathway for large molecule permeation (*Chekeni et al., 2010*; *Chiu et al., 2017*; *Medina et al., 2020*; *Nielsen et al., 2020*; *Poon et al., 2014*). In particular, the dye/metabolite permeation associated with apoptosis requires Panx1 expression, caspase activation, and an intact C-terminal caspase cleavage site; it is blocked by Panx1 channel inhibitors, indicating that ongoing channel activity is required (*Chekeni et al., 2010*; *Chiu et al., 2017*; *Poon et al., 2014*; *Qu et al., 2011*). This dye and metabolite permeation occurs not only during cell death, but is also observed when the channels are activated by C-terminal truncation independent of apoptosis (*Chiu et al., 2017*; *Sandilos et al., 2012*). In addition, ATP release and dye uptake closely parallel the quantized channel activation associated with individual subunit C-tail removal, further implicating the Panx1 channel as the permeation pathway for large molecules (*Chiu et al., 2017*). Despite this substantial body of evidence, this previous work was all based on experiments in intact cells, where secondary release mechanisms could not be excluded, and it has been suggested based on other channel properties (i.e., preferential anion selectivity and single channel conductance) that caspase-cleaved Panx1 is not compatible with large molecule permeation (*Wang and Dahl, 2018*). The present data demonstrate dye and metabolite permeation directly through purified Panx1 channels reconstituted in proteoliposomes to convincingly dispel this conflicting idea and clearly establish that dye/metabolite flux can indeed occur directly through caspase-activated Panx1 channels.

## Channel properties of Panx1 channels in bilayers and cells

Similar to electrophysiological measurements of fPanx1 and hPANX1 recorded in inside-out patches from mammalian cells, we found that fPanx1 was silent in planar lipid bilayers until activated by caspase cleavage. In native cell membranes, caspase-cleaved hPANX1 and fPanx1 generate outwardly-rectifying currents with a single channel conductance of ~90 pS at depolarized potentials (*Chiu et al., 2017*). However, different conductance levels for channels attributed to Panx1 have been observed in various cell systems after different forms of activation, reportedly up to ~500 pS (*Bao et al., 2004*; reviewed in *Chiu et al., 2018*; *Wang et al., 2014*). Here, we observed two main, non-rectifying conductance levels from purified fPanx1 channels in bilayers after caspase activation (~100 pS; ~190 pS). In lipid bilayer recordings of purified hPANX1 reported during preparation of this work, Mou et al. detected a ~ 30 pS channel in *E. coli* polar lipid extract, although these recordings were in the absence of any specific form of activation and were not accompanied by open-closed transitions; the C-tail-cleaved channels, by contrast, yielded a number of extremely large open channel conductance states (i.e., 750 pS, 1.3 nS, and 1.8 nS; at +100 mV) (*Mou et al., 2020*). There have been other reports of constitutive Panx1 channel activity (i.e., in the absence of stimulation), and although these channels display a unitary conductance similar to that of channels stimulated by Casp3 cleavage (~70–80 pS), they do not appear to support ATP release (*Ma et al., 2012*; *Romanov et al., 2012*). It was recently proposed that a second permeation pathway, visible as a side tunnel in the structure of hPANX1, could support atomic ion flux through unstimulated and C-tail-intact channels in the absence of large molecule permeation (*Ruan et al., 2020*); this intriguing idea will need to be explored further (e.g., with single channel recordings of hPANX1 with intact C-termini). At this point, the mechanisms controlling basal and stimulated channel activity, and the factors that determine the vast range of reported Panx1 channel conductance levels, remain to be determined and may be related to any number of cell specific factors (e.g., channel-interacting proteins, different lipid composition in native cell membranes and bilayers).

## A preference for anion versus cation selectivity

In ion substitution experiments, obvious shifts in reversal potential ($E_{rev}$) were observed when $Cl^-$ was exchanged with other negatively charged counterions (*Ma et al., 2012*; *Michalski et al., 2020*; *Romanov et al., 2012*; *Ruan et al., 2020*; *Wang and Dahl, 2018*). These shifts in $E_{rev}$ were strongly affected by mutations in R75, a positively charged residue next to W74, the narrowest pore constriction site identified in multiple recent structural reports (*Deng et al., 2020*; *Michalski et al., 2020*; *Ruan et al., 2020*). These results provide strong evidence for anion permeation through Panx1. By contrast, negligible changes in $E_{rev}$ accompany substitution with different positively charged counterions (*Michalski et al., 2020*; *Ruan et al., 2020*). On this basis, it has been

inferred that the channel is exclusively anion-selective and impermeable to cations (*Ma et al., 2012*). Our data likewise reveal permeation of anionic dyes and signaling molecules (ATP, glutamate), with a clear preference for anionic over cationic dyes. However, we also detected flux through the channel of a positively charged dye (RhB) and metabolite (spermidine), indicating that cationic molecules can also traverse the channel. In addition, we observed simultaneous Panx1-mediated flux of both ATP and spermidine when assessed in combination. This supports the previous observation from apoptotic T cells that multiple important signaling metabolites can be released via caspase-activated channels in a complex milieu containing numerous permeants (*Medina et al., 2020*).

### Permeant size considerations and pore architecture

Our data reveal a negative correlation between dye size and permeation rate – larger dyes were associated with slower permeation. Previous studies have predicted the maximum size of a Panx1 permeant to be less than 1270 Da (7-AAD) (*Chiu et al., 2017*; *Poon et al., 2014*) or 1500 Da (PEG 1500) (*Wang et al., 2007*). We found that Alexa555 (980 Da) could transit via the cleavage-activated Panx1 channel while ATTO550 (1363 Da) was unable to permeate. This confirms that the pore of the activated channel must be able to accommodate a molecule at least up to ~1 kDa.

A major breakthrough in the field was achieved with the recent determination of several high-resolution cryoEM structures of frog and human Panx1 by multiple groups (*Deng et al., 2020*; *Jin et al., 2020*; *Michalski et al., 2020*; *Mou et al., 2020*). The architecture and contours of the transmembrane pore were remarkably similar for all constructs analyzed, despite those including full length (presumably inactive) and C-terminal cleaved/truncated (presumably activated) (*Deng et al., 2020*; *Michalski et al., 2020*; *Ruan et al., 2020*). The narrowest pore constriction was formed by a ring of seven W74 residues on the extracellular surface of the channel, with a diameter of ~8–10 Å. This constriction appears to be too small to accommodate large dyes or even hydrated ATP, although this may be influenced by the shape and flexibility of the permeant molecule and/or the pore itself. Moreover, upon activation by cleavage (or truncation) at the C-terminal caspase site, Panx1 transitions between open and closed states (*Chiu et al., 2017*). Thus, the available structures may have captured the channel in a state that is 'open' for atomic ions but remains 'closed' for large molecules. The possibility of multiple open conformations or separate permeation pathways for ions and large molecules in both Panx1 channels and in CALHM1 channels have been proposed (*Gaete et al., 2020*; *Nielsen et al., 2020*; *Ruan et al., 2020*; *Wang et al., 2014*). If those distinct conformations exist, the dye/metabolite flux we present indicates that Panx1 cleavage by caspase is permissive for conformations that allow large molecule permeation, perhaps by removing a physical block to pore access while allowing open-closed transitions (*Sandilos et al., 2012*).

### Summary

Examination of dye and metabolite flux through purified Panx1 channels reconstituted in proteoliposomes demonstrates directly that molecules up to ~1 kDa are capable of permeating caspase cleavage-activated fPanx1. Moreover, anionic and cationic metabolites and dyes can traverse caspase-activated Panx1 channels, albeit with a marked preference for anionic molecules. These data validate caspase-activated Panx1 as a direct conduit for release of ATP and other signaling molecules that facilitate immunologically silent cell clearance in the context of caspase-dependent cell death (apoptosis, pyroptosis) (*Chekeni et al., 2010*; *Medina et al., 2020*; *Qu et al., 2011*; *Yang et al., 2015*).

## Materials and methods

**Key resources table**

| Reagent type (species) or resource | Designation | Source or reference | Identifiers | Additional information |
|---|---|---|---|---|
| Gene (*Xenopus tropicalis*) | fPanx1 | GenBank | NM_001130256.1 | |

*Continued on next page*

*Continued*

| Reagent type (species) or resource | Designation | Source or reference | Identifiers | Additional information |
|---|---|---|---|---|
| Gene (*Homo sapiens*) | hPANX1 | GenBank | NM_015368.4 | |
| Cell line (*Spodoptera frugiperda*) | Sf9 | Expression Systems, Davis, CA | Parent cell line: IPLB-Sf-21-AE, RRID:CVCL_0518 | Clonal isolate derived from the parental cell line |
| Cell line (*Homo sapiens*) | HEK293T | ATCC | CRL-3216, RRID:CVCL_0063 | Negative for mycoplasma contamination at ATCC (obtained from Kodi Ravichandran lab) |
| Biological sample (*Homo sapiens*) | Caspase-3 | Kang et al. | | |
| Antibody | Anti-human/mouse/rat Pannexin-1 (rabbit monoclonal) | Cell Signaling Technology | D9M1C, RRID:AB_28000167 | 1:1000 dilution |
| Chemical compound, drug | Brain PC | Avanti Polar Lipids | 840053 | |
| Chemical compound, drug | Brain Total Lipid Extract | Avanti Polar Lipids | 131101 | |
| Chemical compound, drug | Brain PI(4,5)P2 | Avanti Polar Lipids | 840046 | |
| Chemical compound, drug | DPhPC | Avanti Polar Lipids | 860337 | |
| Chemical compound, drug | Sulforhodamine-B | Sigma Aldrich | S1402 | |
| Chemical compound, drug | Rhodamine B | Sigma Aldrich | 83689 | |
| Chemical compound, drug | Dextran, Tetramethylrhodamine, 3000 MW, | ThermoFisher Scientific | D3307 | |
| Chemical compound, drug | Alexa 594 Carboxylic Acid, tris salt | ThermoFisher Scientific | A33082 | |
| Chemical compound, drug | Alexa 555 Carboxylic Acid, tris salt | ThermoFisher Scientific | A33080 | |
| Chemical compound, drug | ATTO 550 Phallodin | ATTO-TEC | AD 550–8 | |
| Chemical compound, drug | $\alpha$-$^{32}$P-ATP | PerkinElmer | BLU003H250UC | |
| Chemical compound, drug | $^{3}$H-Glutamate | PerkinElmer | NET490250UC | |
| Chemical compound, drug | $^{3}$H-Spermidine | PerkinElmer | NET522001MC | |

*Continued on next page*

*Continued*

| Reagent type (species) or resource | Designation | Source or reference | Identifiers | Additional information |
| --- | --- | --- | --- | --- |
| Software, algorithm | LabView | Kiessling et al. | | https://github. com/VolkerKirchheim/ VK_TIRFsinglevesicle Step1 ; *Narahari, 2021*; (copy archived at 'swh:1:rev:79a55c09884d1fd3f4965ef1d1bf8f102711d828') |

## Panx1 expression

A cDNA encoding *Xenopus tropicalis* Panx1 (fPanx1; Genscript-OXa25378, Accession NM_001130256.1) was inserted upstream of a thrombin protease cleavable linker (LVPRGS), enhanced green fluorescent protein (eGFP), and a Strep II epitope (WSHPQFEK) in a modified pFast-BacI vector (Invitrogen, Carlsbad, CA) by In-Fusion cloning (Takara Bio USA, Inc, Mountain View, CA). Briefly, a PCR amplicon containing fPANX1 with a 5' NotI and XhoI site at its 3' end was prepared using Pfu DNA polymerase and inserted into a modified pFastBacI vector that contained a unique XhoI site upstream of the thrombin proteolytic site. The Strep II tag was fused in frame to the C-terminus of eGFP. The construct was verified by DNA sequencing.

The Bac-to-Bac expression system (Invitrogen, Carlsbad, CA) was used to generate baculovirus for expression in *Spodoptera frugiperda* (Sf9) insect cells. Recombinant fPanx1-eGFP baculovirus was used to infect Sf9 insect cells grown at 27°C to a density of $2 \times 10^6$ mL$^{-1}$, at a multiplicity of infection (MOI) of ~3. Cells were collected 48 hr after infection by low-speed centrifugation at 2000 × g and stored at −80°C.

## Panx1 protein purification

To isolate membrane-localized fPanx1-eGFP, Sf9 cell pellets were resuspended in low salt buffer (50 mM HEPES, pH 7.5, 50 mM NaCl, 0.5 mM EDTA, with protease inhibitor cocktails (Roche, Basel Switzerland) and lysed by Dounce homogenization (~30 strokes). Nucleic acids were digested by adding 2.5 mM MgCl$_2$ and ~12.5 units of Benzonase (EMD Millipore, Burlington MA) per 1 mL lysate, with gentle stirring at 4°C for 10 min. Membranes were collected by ultracentrifugation at 100,000 × g and washed with stepwise Dounce homogenization again in low salt buffer and twice in high salt buffer (50 mM HEPES, pH 7.5, 1 M NaCl, 0.5 mM EDTA, with protease inhibitor cocktails). Pellets were isolated by ultracentrifugation at 100,000 × g between steps. Finally, membranes were resuspended (5 mL/g) in membrane freezing buffer (10 mM HEPES, pH 7.5, 20 mM KCl, 10 mM MgCl$_2$, and 40% glycerol), flash frozen, and stored at −80°C.

The frozen membrane pellet was thawed and solubilized at 4°C for 3 hr with 1% (w/v) n-dodecyl-β-D-maltopyranoside (DDM; Anatrace, Maumee, OH) and 0.2% (w/v) cholesteryl hemisuccinate (CHS; Anatrace, Maumee, OH) in ~50 mL of buffer containing 50 mM HEPES, pH 7.5, 300 mM NaCl, 3 mM CaCl$_2$, 2.5% glycerol and protease inhibitor cocktails. Insoluble material was removed by ultracentrifugation at 100,000 × g, and the supernatant was incubated with ~1.0 mL of Strep-Tactin Superflow Plus resin (QIAGEN, Hilden, Germany) overnight at 4°C. The resin was packed in an Econo-column (1.0 × 10 cm; Bio-Rad, Hercules, CA) and washed with low salt buffer (50 mM HEPES, pH 7.5, 300 mM NaCl, 3 mM CaCl$_2$, and 0.2% DDM with 0.04% CHS) for 20 column volumes/wash, high salt buffer (50 mM HEPES, pH 7.5, 1 M NaCl, 3 mM CaCl$_2$, and 0.05% DDM with 0.01% CHS), for 20 column volumes/wash, and eluted with 2.5 mM Desthiobiotin (Sigma-Aldrich, St. Louis, MO) in buffer (50 mM HEPES, pH 7.5, 500 mM NaCl, 3 mM CaCl$_2$, and 0.02% DDM with 0.004% CHS). The eluted protein was concentrated to ~500 µL using an 100 kDa Amicon ultracel-100 centrifugal filter unit (EMD Millipore, Burlington MA). Preparative SEC was performed on a Superose 6 Increase 10/300 GL column (GE Healthcare, Chicago, IL) interfaced to an AKTA Purifier 10 FPLC system (GE Healthcare, Chicago, IL), equilibrated with buffer (50 mM HEPES, pH 7.5, 500 mM NaCl, 3 mM CaCl$_2$, and 0.02% DDM with 0.004% CHS). Fractions containing fPanx1-eGFP were collected and the protein was concentrated to ~2–3 mg mL$^{-1}$ using a 100 kDa Amicon ultracel-100 centrifugal filter unit and stored at 4°C for proteoliposome reconstitution, or snap frozen and stored at −80°C for lipid bilayer recordings.

## Liposome preparation and reconstitution of Panx1

Liposomes were prepared from lipids of the following composition: 70% brain phosphatidylcholine (PC), 15% total brain lipid extract, 14% cholesterol, 1% phosphatidylinositol 4,5-bisphosphate ($PIP_2$; all from Avanti Polar Lipids, Alabaster, AL), with 40 µL of methanol, dried under nitrogen flow with gentle agitation and then in a vacuum desiccator overnight. Dried lipids were resuspended in liposome buffer (20 mM Tris, 140 mM NaCl, pH 7.4), vortexed vigorously, incubated at room temperature for 45 min, and the lipid mixture was extruded at least 21 times through a LiposoFast-Basic extruder (Avestin Inc, Ottawa, Canada) with a 100 nm polycarbonate membrane. After extrusion, sodium cholate (3.9 mM final concentration) was added, and the solution was gently agitated for 3 hr. Purified detergent-solubilized fPanx1-eGFP (21.6 µM) was added in a 1:1000 protein:lipid molar ratio and gently agitated for 1 hr at room temperature. Following protein incubation, the proteoliposome-containing solution was transferred to 10 kDa dialysis cassettes (Slide-A-Lyzer; Thermo Scientific, Waltham, MA) and immersed in liposome buffer containing BioBeads (Bio-Rad, Hercules, CA) for detergent removal (5 L at 4°C, with gentle stirring), first for 5 hr and then overnight at 4°C with fresh BioBead-containing liposome buffer. Proteoliposomes were collected by $100,000 \times g$ spin for 60 min at 4°C and resuspended to 1 mL with liposome buffer and utilized immediately or snap frozen in liposome buffer containing 200 mM sucrose. After snap freezing in liquid $N_2$, liposomes were stored at −80°C until use. For dye uptake experiments, proteoliposomes were thawed at 4°C, diluted $10\times$ in liposome buffer, spun at $100,000 \times g$, and resuspended to desired volume.

Liposomes for radiolabeled metabolite uptake were prepared as previously described (*Johnson and Lee, 2015*). Briefly, the dried lipid mixture described above was resuspended in liposome buffer (20 mM Tris, 140 mM NaCl, pH 7.4), incubated at room temperature for 45 min, vortexed vigorously, and sonicated in cycles of 1 min in a bath sonicator followed by 1 min on ice until the solution was clear. Liposomes were incubated with 3.9 mM Na Cholate (Sigma-Aldrich, St. Louis, MO) and rotated for 1 hr under nitrogen. After addition of fPanx1-eGFP (1:1000 protein:lipid molar ratio), the mixture was rotated for 2–3 hr at room temperature under nitrogen. The proteoliposome-containing solution was transferred to 10 kDa dialysis cassettes (Slide-A-Lyzer; Thermo Scientific, Waltham, MA) and immersed in liposome buffer containing BioBeads for detergent removal (5 L at 4°C, with gentle stirring), first for 5 hr and then overnight at 4°C with fresh BioBead-containing liposome buffer. Proteoliposomes were snap frozen in liquid $N_2$ and stored at −80°C until use.

## Nycodenz cofloatation assay

Nycodenz cofloatation was performed as previously described (*Hernandez et al., 2012*). Briefly, 50 µL proteoliposomes were mixed with 50 µL 80% w/v Nycodenz in liposome buffer. A 50 µL layer of 30% w/v Nycodez solution was applied on top of the liposome-Nycodenz mixture, and an additional 50 µL of liposome buffer layered on top. The density gradient was spun at 197,000 g for 90 min at 4°C in a TL-100 ultracentrifuge (TLS55 Rotor; Beckman-Coulter, Brea, CA). Upon completion, 20 µL fractions were collected and analyzed via SDS PAGE and silver staining.

## Gel electrophoresis and silver staining

SDS PAGE gels (BioRad AnyKD Mini-PROTEAN TGX; BioRad, Hercules, CA) were run at 120 mV for 45 min and gels were fixed in 50 mL of 40% methanol solution containing 0.0185% formaldehyde, washed in DI water (2 × 5 min), immersed in 0.02% sodium thiosulfate solution (1 min), and washed again with in DI water (2 × 20 s). The gel was incubated in 50 mL of 0.1% silver nitrate solution for 10 min. The gel was quickly washed with 10 mL of DI water and then washed with 10 mL of thiosulfate developing solution (0.0185% formaldehyde, 28.3 mM sodium carbonate, and 0.0004% sodium thiosulfate). The gel was incubated in 50 mL of thiosulfate developing solution until bands were visualized. A volume of 2.5 mL of 2.3 M citric acid was added for 10 min to stop developing. The gel was washed with water and imaged.

## Negative stain electron microscopy

For negative staining, 3.5 µL of proteoliposomes were applied to a glow-discharged, carbon-coated, 300-mesh, copper grid (Electron Microscopy Sciences, Hatfield, PA) and stained with 2% uranyl acetate (*Adair and Yeager, 2007*). Low-dose EM was performed at the Molecular Electron Microscopy Core facility at UVA using a Tecnai F20 electron microscope (FEI, Hillsboro, OR), operating at 120

kV. Images were recorded at a nominal magnification of 29,000× and a defocus of 3 μm using a 4 × 4 K charge-coupled device camera (UltraScan 4000; Gatan, Pleasanton, CA), corresponding to a pixel size of 3.7 Å on the specimen. Proteoliposome diameters were obtained using ImageJ (NIH, Bethesda, MD) from the average of two measurements per proteoliposome.

## Casp3 purification

Recombinant Casp3 precursors were prepared as previously described (*Kang et al., 2008*). In brief, BL21(DE3) cells were transformed with the pro-Casp3 Δ28/175TS deletion in pET-22b (+) vector and treated with 1 mM IPTG (18°C, 18 hr). Cells were lysed using a microfluidizer, pro-Casp3 precursors were purified, and then activated by thrombin as previously described (*Kang et al., 2008*).

## Western blotting

Proteoliposomes (25 μL, 1:1000 protein:lipid ratio) were incubated with purified Casp3 (6 μL, $k_{cat}$ = ~1.9 ± 0.1) and analyzed via SDS PAGE electrophoresis (BioRad AnyKD Mini-PROTEAN TGX; Bio-Rad, Hercules, CA). Samples were transferred to 0.45 μM nitrocellulose membranes (Perkin Elmer), which were blocked for 1 hr at room temperature in 5% non-fat milk, 10 mM Tris, 150 mM NaCl, 0.1% Tween 20, pH 7.4 and then incubated overnight at 4°C with fPanx1 antibody (Rabbit mAb #91137, 1:1000; Cell Signaling Technology, Danvers, MA). After three washes in a Tris-based buffer (10 mM Tris, 150 mM NaCl, 0.1% Tween 20, pH 7.4), the membranes were incubated with horseradish peroxidase-conjugated secondary antibody (Na9340; 1:10000; Amersham, Little Chalfont, UK), and immunoreactive signals were detected by enhanced chemiluminescence (Western Lightning Plus-ECL; PerkinElmer, Waltham, MA) and visualized using Amersham Hyperfilm ECL (GE Healthcare, Chicago, IL).

## Planar lipid bilayer recordings

Single-channel activity was evaluated in planar lipid bilayers using the Orbit mini system (Nanion Technologies, Munich, Germany). Briefly, Multi Electrode Cavity Array (MECA4) chips (Ionera, Freiburg, Germany) were filled with 150 μL of solution containing 200 mM KCl, 5 mM HEPES, and 0.2 mM EDTA; adjusted to pH 7.6. Lipid bilayers were formed by painting the chips with 10 mg/mL 1,2-diphytanoyl-sn-glycero-3-phosphocholine (DPhPC; Avanti Polar Lipids, Alabaster, AL) dissolved in octane. Purified fPanx1-eGFP (2–5 μL) was added at the cis (ground) side of the bilayer. After channel reconstitution, the current was recorded in a range of voltages (±200 mV). Activity at extreme potentials (>140 mV) was indicative of channel presence in the bilayer. Then, 2 μL of Casp3 (~0.4 mg/mL) was added to the cis side of the bilayer. After channel activation, Casp3 was gently washed out to keep symmetrical ionic composition on both sides of the bilayer. After treatment with caspase, the current was recorded again in a range of voltages (±200 mV). Recordings were performed at 20 kHz using Element Data Recorder 3.8.0 software and further analyzed with Clampfit 10 software (Axon Instruments, San Jose, CA). Measurements were performed at 37°C using a temperature control unit (Nanion Technologies, Munich, Germany).

## Whole cell recordings

Whole cell voltage clamp recordings of fPanx1 and hPANX1 were performed in transiently transfected HEK293T cells (ATCC, Manassas, VA. Cells were authenticated originally by ATCC STR profiling and negative for mycoplasma at time of purchase), as described previously (*Chiu et al., 2017*). In short, expression plasmids for fPanx1-eGFP or hPANX-TEV and TEV protease (1:3) (*Sandilos et al., 2012*) were transfected into HEK293T cells using Lipofectamine 2000 (Thermo Fisher Scientific, Waltham, MA). After 16–18 hr, whole cell recordings were performed at room temperature using borosilicate glass micropipettes (Harvard Apparatus, Holliston, MA) that were pulled on a P-97 puller (Sutter Instrument Company, Novato, CA) to a resistance of 3–5 MΩ and coated with Sylgard 184 (Dow Corning Corporation, Midland, MI). Recordings were obtained with an Axopatch 200B amplifier, a Digidata 1322 A board, and Clampex software (all Molecular Devices, San Jose, CA) with a HEPES-bath solution composed of (mM): 140 NaCl, 3 KCl, 2 MgCl₂, 2 CaCl₂, 10 HEPES, and 10 glucose (pH 7.3) and an internal solution composed of (mM): 100 CsMeSO₄, 30 TEACl, 4 NaCl, 1 MgCl₂, 10 HEPES, 10 EGTA, 3 ATP-Mg, and 0.3 GTP-Tris (pH 7.3). Purified and activated Casp3 was added in the internal solution (2 μg/mL) to cleave and activate fPanx1-eGFP before bath application

of CBX (50 µM). CBX-sensitive currents from fPanx1-eGFP or hPANX-TEV were obtained from ramp voltage commands, and normalized to the peak current to compare current–voltage relationships of fPanx1-eGFP and hPANX-TEV.

## Single channel recordings

We examined single channel activity of caspase-activated fPanx1-GFP in transfected HEK293T cells (as above). For inside-out patch recordings, Sylgard-coated patch pipettes were pulled to a DC resistance of 7–10 MΩ and filled with the HEPES-bath solution. After seal formation ($\geq$10 GΩ) and patch excision, the bath solution was exchanged to an inside-out solution containing 150 mM CsCl, 5 mM EGTA, 10 mM HEPES, and 1 mM $MgCl_2$ (pH 7.3). Patches were held at +50 to +80 mV ($\Delta$ 10 mV), and only those patches that were silent initially after excision were used (i.e., those without native channel activities). Purified Casp3 was applied in the proximity of the patch to a final bath concentration of ~1–2 µg/mL. Stable steady-state channel activity was recorded ~5–10 min after Casp3 addition before switching to an inside-out bath solution containing CBX (50 µM). Channel amplitudes were obtained from all-points amplitude histograms at multiple patch potentials using Clampfit 10 (Molecular Devices).

## ImageStream flow cytometry

Proteoliposomes (50 µL) were incubated in liposome buffer with Casp3 (75 µg/mL final) overnight at 4°C. SR-B dye was added to the reaction and incubated for 3 hr. The reaction was diluted to obtain a final SR-B concentration of 100 µM prior to flow cytometry analysis. The entire mixture was loaded on to an Amnis ImageStream MkII imaging flow cytometer (Luminex Corporation, Austin, TX). The 488 nm laser was used to capture eGFP signal and the 560 nm laser was used to capture Sulforhodamine-B. Amnis IDEAS was utilized for data processing. A decrease in GFP fluorescence was used to verify caspase-mediated fPanx1-GFP cleavage for experiments included in the statistical analysis.

## Dye loading into proteoliposomes

Proteoliposomes were filled with dye as previously described (*Karasawa et al., 2017*). Briefly, fPanx1-containing proteoliposomes or empty liposomes (i.e., without fPanx1) were incubated with each dye: Sulforhodamine B, Rhodamine B, (all 1 mM; Sigma Aldrich, St. Louis, MO); Alexa 594 Carboxylic Acid, Alexa 555 Carboxylic Acid, Dextran Tetramethylrhodamine 3000 MW (all 1 mM; Thermo Fisher Scientific, Waltham, MA); ATTO 550 Phalloidin (25 µM ATTO-TEC, Siegen, Germany), and loaded by three sequential freeze-thaw cycles in liquid $N_2$ and room temperature water bath. After the final thaw, the mixture was extruded through a 100 nm extruder (T and T Scientific, Knoxville, TN). The proteoliposomes containing dye were eluted through a GE Healthcare PD MiniTrap column containing G-25 resin equilibrated with liposome buffer (GE Healthcare, Chicago, IL). The eluant was spun at 100,000 × g for 1 hr at 4°C. The proteoliposome/liposome pellet filled with dye was resuspended with cold liposome buffer.

## TIRF microscopy and data analysis of TIRF

A Zeiss AxioObserver Z1 fluorescence microscope (Carl Zeiss, Oberkochen, Germany) with a 63× water immersion objective (N.A. = 0.95) and a prism-based illumination was utilized. The light sources for excitation were an OBIS 488 LX and an OBIS 561 LS laser (Coherent Inc, Santa Clara, CA). An OptoSplit (Andor, Belfast, Northern Ireland) was used to separate two spectral fluorescence bands with band pass filters BP525/50 and BP607/70 (Idex-Semrock, Rochester, NY). Images were acquired every 10 s with an EMCCD iXon DV887ESC-BV (Andor). Laser intensity, shutter, and camera were controlled by homemade software written in LabVIEW (National Instruments, Austin, TX). Custom chambers were filled with liposomes (1:1000 liposome:buffer dilution) and injected with Casp3 diluted in liposome buffer (to ~0.01 mg/mL).

Image series were analyzed by extracting the central pixel values from both spectral channels from regions of interests around each observed liposome by custom software written in LabVIEW (National Instruments; software has been made available: https://github.com/VolkerKirchheim/VK_TIRFsinglevesicleStep1) (copy archived at https://archive.softwareheritage.org/swh:1:rev:79a55c09884d1fd3f4965ef1d1bf8f102711d828/) (*Kiessling, 2020*; *Kiessling et al., 2006*). Efflux amounts were quantified by measuring the decrease of fluorescence intensity relative to the intensity

before the onset of the decay and the background. Flux rates were determined by fitting mono exponential decay curves to the data starting at the onset of the decay over a range of 1200 s.

## Bulk dye uptake assay

Proteoliposomes (50 µL) were incubated overnight with Casp3 (75 µg/mL) at 4°C in liposome buffer (20 mM Tris pH 7.4, 140 mM NaCl). Dye (1 mM SR-B) was added to the mixture (final volume 100 µL) and gently agitated at 4°C for 3 hr. CBX in liposome buffer was added to the liposome reaction (50 µM, in 200 µL final volume). The reaction was gently layered onto GE Healthcare PD MiniTrap columns containing G-25 resin pre-equilibrated with liposome buffer containing 100 µM CBX (GE Healthcare, Chicago, IL). The column was spun at $1000 \times g$ for 1 min at 4°C. The eluted liposomes were read on a FlexStation 3 Multi-Mode Microplate Reader (excitation 565 nm, emission 586 nm) (Molecular Devices, San Jose, CA).

## Metabolite uptake assay

As previously described (*Johnson and Lee, 2015*), frozen liposomes were thawed on ice and subjected to three freeze/thaw cycles using liquid $N_2$ and a room temperature water bath. Following the third freeze/thaw cycle, liposomes were extruded through 1.0 µm polycarbonate membranes in a Mini-Extruder (Avanti Polar Lipids Inc, Alabaster, AL). Proteoliposomes (100 µL) were incubated overnight with Casp3 (~75 µg/mL) at 4°C in liposome buffer (20 mM Tris pH 7.4, 140 mM NaCl). We examined uptake of each of the potential permeants independently. For this, metabolites were added to caspase-treated proteoliposome mixtures to final concentrations of 1 mM ATP, 0.8 mM glutamate, 8 µM spermidine, together with 0.02 µCi/µL of $^{32}$P-ATP, $^3$H-Glutamate or $^3$H-Spermidine (200 µL final volume; all from PerkinElmer, Waltham, MA). Note that we used a 100-fold lower final concentration of spermidine to avoid potential disruption of the proteoliposomes (*Creutz et al., 2012*) and thus the hot:cold molar ratio for spermidine was ~100-fold greater than for glutamate. Following gentle agitation at room temperature for 3 hr, the radioactive mixture was diluted with ice cold liposome buffer containing unlabeled metabolite at the relevant final concentrations (to 1 mL) and filtered through a Whatman GF/B filter pre-equilibrated with liposome buffer containing unlabeled metabolite. The filter was washed three times with ice cold liposome buffer containing unlabeled metabolite, immersed in 5 mL of Ecoscint A (National Diagnostics, Atlanta, GA), and read with a LS6500 Multipurpose Scintillation Counter (Beckman Coulter, Brea, CA).

## Chemical structures

All chemical structures were rendered in ChemDraw (Cambridgesoft, Cambridge, MA) with the base structures (carboxylic acids of Alexa 594 and Alexa 555) and the dextran conjugate of tetramethylrhodamine (Thermo Fisher Scientific; see *Gebhardt et al., 2020*); the precise site of the conjugate for these proprietary dyes was not provided and so the figures depict approximate structures. The chemical structure of ATTO 550 Phallodin was provided by ATTO-TEC.

## Acknowledgements

This work was supported by P01 HL120840 (KSR, DAB, MY); R01 HL138241 (PQB); R01 GM099490 (JEC); R01 GM138532 (MY); and R01 HL48908 (MY); and P01 GM072694 (LKT) and R01 GM051329 (LKT). AKN was supported by F30 CA236370, T32 GM007267, and the University of Virginia Whitfield-Randolph Scholarship, and AKN and CBM were supported by T32 GM007055. YHC was supported by MOST 108–2320-B-007–007-MY2. The authors are grateful for sample preparation by Sandra Poulos, liposome preparation advice from Raghavendar Reddy Sanganna Gari and Patrick Seelheim, helpful early advice on liposome bulk dye uptake assays from Joseph A Mindell (NINDS), and the University of Virginia flow cytometry core facility. The authors are grateful to Elizabeth Gonye, Keyong Li, and Yingtang Shi (Bayliss Laboratory) for helpful discussions and suggestions.

# Additional information

## Funding

| Funder | Grant reference number | Author |
|--------|------------------------|--------|
| National Institutes of Health | P01 HL120840 | Kodi S Ravichandran<br>Mark Yeager<br>Douglas A Bayliss |
| National Institutes of Health | R01 HL138241 | Paula Q Barrett |
| National Institutes of Health | R01 GM099490 | Jorge E Contreras |
| National Institutes of Health | R01 HL48908 | Mark Yeager |
| National Institutes of Health | R01 GM138532 | Mark Yeager |
| National Institutes of Health | P01 GM072694 | Lukas K Tamm |
| National Institutes of Health | R01 GM051329 | Lukas K Tamm |
| National Institutes of Health | F30 CA236370 | Adishesh K Narahari |
| National Institutes of Health | T32 GM007055 | Adishesh K Narahari<br>Christopher B Medina |
| National Institutes of Health | T32 GM007267 | Adishesh K Narahari |
| University of Virginia | Whitfield-Randolph Scholarship | Adishesh K Narahari |
| Ministry of Science and Technology, Taiwan | 108-2320-B-007-007-MY2 | Yu-Hsin Chiu |

The funders had no role in study design, data collection and interpretation, or the decision to submit the work for publication.

## Author contributions

Adishesh K Narahari, Conceptualization, Formal analysis, Investigation, Visualization, Methodology, Writing - original draft, Project administration, Writing - review and editing; Alex JB Kreutzberger, Conceptualization, Formal analysis, Investigation, Methodology, Writing - review and editing; Pablo S Gaete, Christopher B Medina, Formal analysis, Investigation, Methodology, Writing - review and editing; Yu-Hsin Chiu, Conceptualization, Formal analysis, Investigation, Writing - review and editing; Susan A Leonhardt, Xueyao Jin, Investigation, Methodology, Writing - review and editing; Patrycja W Oleniacz, Conceptualization, Software, Investigation, Methodology, Writing - review and editing; Volker Kiessling, Conceptualization, Resources, Software, Investigation, Methodology, Writing - review and editing; Paula Q Barrett, Resources, Supervision, Funding acquisition, Methodology, Writing - review and editing; Kodi S Ravichandran, Mark Yeager, Jorge E Contreras, Resources, Supervision, Funding acquisition, Writing - review and editing; Lukas K Tamm, Conceptualization, Resources, Supervision, Funding acquisition, Visualization, Writing - original draft, Project administration, Writing - review and editing; Douglas A Bayliss, Conceptualization, Resources, Supervision, Funding acquisition, Investigation, Visualization, Methodology, Writing - original draft, Project administration, Writing - review and editing

## Author ORCIDs

Adishesh K Narahari (iD) https://orcid.org/0000-0002-8708-9161
Alex JB Kreutzberger (iD) https://orcid.org/0000-0002-9774-115X
Pablo S Gaete (iD) http://orcid.org/0000-0003-3373-9138
Yu-Hsin Chiu (iD) http://orcid.org/0000-0002-4730-8104
Volker Kiessling (iD) http://orcid.org/0000-0002-9388-5703
Jorge E Contreras (iD) http://orcid.org/0000-0001-9203-1602
Lukas K Tamm (iD) http://orcid.org/0000-0002-1674-4464
Douglas A Bayliss (iD) https://orcid.org/0000-0002-5630-2572

## Decision letter and Author response

Decision letter https://doi.org/10.7554/eLife.64787.sa1
Author response https://doi.org/10.7554/eLife.64787.sa2

## Additional files

### Supplementary files
• Transparent reporting form

### Data availability

All data generated or analyzed during this study are included in the manuscript and supporting files (Source Data files). Source data files have been provided for Figure 1G, Figure 1 Supplement 1C, Figure 1 Supplement 2B-E, Figure 2D, Figure 2 Supplement 1, Figure 2 Supplement 2A-F, Figure 3B-I, Figure 4 B-D, Figure 4 Supplement 2B-C. Source code has been uploaded to Github: https://github.com/VolkerKirchheim/VK_TIRFsinglevesicleStep1. (Copy archived at https://archive.software-heritage.org/swh:1:rev:79a55c09884d1fd3f4965ef1d1bf8f102711d828/).

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
