## [Decision Letter]

**Acceptance summary:**

Pannexins are membrane channels that are found in many cell types. It has been suggested that these channels when activated can transport ATP and other bulky metabolites. However, those studies cannot rule out the possibility that these ions flux through secondary pathways that are stimulated by the channel. Here, using a purified reconstituted system, the authors provide the most definitive evidence that Pannexin 1 when activated by caspase can flux large bulky metabolites such as ATP and spermidine. This study resolves a key question and has important physiological significance.

**Decision letter after peer review:**

Thank you for submitting your article "Caspase-activated Pannexin 1 channels favor anionic molecules and support flux of ATP and other signaling metabolites" for consideration by *eLife*. Your article has been reviewed by three peer reviewers, including Baron Chanda as the Reviewing Editor and Reviewer #1, and the evaluation has been overseen by a Reviewing Editor and Kenton Swartz as the Senior Editor.

The reviewers have discussed the reviews with one another and the Reviewing Editor has drafted this decision to help you prepare a revised submission.

Summary:

The manuscript by Narahari et al., reports comprehensive assessment of permeation properties of the frog Panx1 channel using liposomes reconstituted with purified fPanx1 proteins. In the past, much had been done using cell-based assay, however, there has not been any liposome based assays with purified protein. Such assays provide definitive characterization of the intrinsic functional properties of the protein. Here, the authors first conducted a lipid-bilayer electrophysiology by fusing the fPanx1 proteolipomes and clearly recapitulated caspase-3 cleavage dependent current formation. They also detected influxing of fluorescent dyes into the proteoliposomes and effluxing of fluorescent dyes from liposomes packaged from the proteoliposomes by TIRF microscopy (measurement of fluorescence decay from single proteoliposomes). The quality control of the experiments is excellent (e.g. verification of caspase cleavage, protein quality, detection of reconstitution, size estimation of proteoliposomes under electron microscopy, etc). This work represents the first solid case where Panx1 activity can be clearly measured using an in vitro liposome assay. Using these established assays, the authors assessed permeation of dyes with different sizes to measure pore sizes. Furthermore, the authors assessed uptake of tritiated ATP, glutamate, and spermidine to show that both cationic and anionic metabolites can permeate with higher selectivity to anionic metabolites. This work clearly addresses a key outstanding question in the field.

Essential revisions:

The authors have placed an excessive emphasis on differences in permeability of cationic vs. anionic molecules by including this in the titles, since only one direct quantitative comparison was made. The observed differences between SR-B over positively charged RhB, while consistent with small anion selectivity assessed previously by electrophysiology, represent only a partial characterization and might be also reflect other reasons. This can be easily addressed by rewording the title. Please see our suggestions below.

1) In Figure 1B, it is not clear from the single channel traces where the Closed state is in some of the traces in the right side. O1 is lined up right along with C in -80 and -100 mV traces.

2) Figure 1G, the inward and outward conductance appear displaced relative to each other almost implying that they have different selectivity. I am assuming that there is a trivial reason I am overlooking.

3) Figure 4. Related to the above point, it would be good for readers to have a sense of how much of the tritiated metabolites actually influxed into the proteoliposomes. For that reason, it is perhaps better to have Y-axis converted to “mole” not CPM.

4) Introduction. Citation for the first CALHM1 structure, Syrjanen,. Furukawa, (2020), and LRRC8 structure, Deneka et al., (2018), should be cited here.

5) Subsection “Preparation of Pannexin 1 Proteoliposomes”. Sf9 only minimally contain cholesterol/sterol. Could that affect functions of the purified Panx1? Alternatively, could the addition of cholesterol change conductance?

6) In Figure 2D, comparison between the left and right panels is not simple. Can the authors indicate studies in the same sets of vesicles using similar colors? Can they generate a graph that shows a comparison of the changes of GFP and SRB for each set?

7) Have the authors attempted an experiment like those in Figure 4 including both ATP and spermidine together (and separately counting 3H and 32P) to provide a direct comparison of the permeabilities of these two molecules?

8) It would be nice if the authors included a short section in the Discussion considering why other investigators have previously found some permeation through un-cleaved Panx1 channels.

9) Channel permeation does not depend only on the charge and mass of the permeant but also on its shape (second largest diameter). The authors might include a supplemental figure showing images of the size and shapes of the different permeants that they have tested. It could be briefly addressed in the Discussion.

10) Title: How about "ATP and bulky signaling metabolites can flux through Caspase activated Pannexin 1 channels" or something along those lines.

---

## [Author Response]

Essential revisions:The authors have placed an excessive emphasis on differences in permeability of cationic vs. anionic molecules by including this in the titles, since only one direct quantitative comparison was made. The observed differences between SR-B over positively charged RhB, while consistent with small anion selectivity assessed previously by electrophysiology, represent only a partial characterization and might be also reflect other reasons. This can be easily addressed by rewording the title. Please see our suggestions below.

We have re-worded the title according to the suggestion provided further below.

1) In Figure 1B, it is not clear from the single channel traces where the Closed state is in some of the traces in the right side. O1 is lined up right along with C in -80 and -100 mV traces.

Thank you for pointing out this oversight. The closed state for the recordings of caspase-treated Panx1 are now presented in Figure 1—figure supplement 3B.

2) Figure 1G, the inward and outward conductance appear displaced relative to each other almost implying that they have different selectivity. I am assuming that there is a trivial reason I am overlooking.

We have refitted the data across negative and positive potentials such that the line describing the slope conductance now passes through the origin. We have no reason to believe that the purified channel displays differential selectivity at positive or negative potentials.

3) Figure 4. Related to the above point, it would be good for readers to have a sense of how much of the tritiated metabolites actually influxed into the proteoliposomes. For that reason, it is perhaps better to have Y-axis converted to “mole” not CPM.

The Y-axis has now been converted to pmoles and not CPM. We also note in the figure legend that molar quantities should not be compared across compounds due to the different assay conditions. That is, we used 4 μCi of each metabolite in the assays, and thus the final concentrations and hot:cold ratios were different. In addition, we used spermidine at much lower final concentration due to its “detergent” properties (see Materials and methods).

4) Introduction. Citation for the first CALHM1 structure, Syrjanen,. Furukawa, (2020), and LRRC8 structure, Deneka et al., (2018), should be cited here.

These references have now been added.

5) Subsection “Preparation of Pannexin 1 Proteoliposomes”. Sf9 only minimally contain cholesterol/sterol. Could that affect functions of the purified Panx1? Alternatively, could the addition of cholesterol change conductance?

We reconstituted purified Panx1 into either cholesterol-containing proteoliposomes for flux assays or cholesterol-free lipid bilayers for planar lipid bilayer recordings.

Cholesterol/sterols can modulate channel activity in numerous ion channel families (Levitan, Singh and Rosenhouse-Dantsker, 2014), and thus it is possible that the absence of cholesterol could affect Panx1 channel activity in the bilayer recordings. The only other lipid bilayer recordings of the purified channel were obtained with hPANX1 in *E. coli* polar lipid extract (from Avanti), which is also devoid of cholesterol (Mou et al., 2020). In that case, Mou et al., recorded multiple single channel current amplitudes from caspase-cleaved hPANX1 channels that are much larger (even up to 1.8 nS) than we see in bilayers or cells (~100 pS). For our bilayer recordings, we also find an additional larger conductance level (~190 pS). At this point, the various factors that can influence these channel recordings, perhaps including cholesterol, remain to be determined.

We now specifically mention that the non-physiological lipids used for our bilayer recordings did not include cholesterol (or PIP2), and that this could have contributed to variations in channel properties (subsection “Pannexin 1 forms a dye permeable pore”). We also now note the composition of lipids in the proteoliposomes at the point where the flux assays are first described (subsection “Pannexin 1 forms a dye permeable pore”).

6) In Figure 2D, comparison between the left and right panels is not simple. Can the authors indicate studies in the same sets of vesicles using similar colors? Can they generate a graph that shows a comparison of the changes of GFP and SRB for each set?

We have now updated Figure 2D to be color coded by experiment. We have also added a supplemental graph (Figure 2—figure supplement 2A) that shows the changes in GFP and SRB for each of the experiments on a single graph. Finally, we have included the corresponding flow cytometry-style dot plots to show the caspase-evoked changes in GFP and SR-B for each of the individual experiments (Figure 2—figure supplement 2B-F).

7) Have the authors attempted an experiment like those in Figure 4 including both ATP and spermidine together (and separately counting 3H and 32P) to provide a direct comparison of the permeabilities of these two molecules?

Thank you for the suggestion. We have now performed experiments to measure ATP and spermidine flux simultaneously in the same experiment (Figure 4—figure supplement 2). In this combined experiment, we find uptake of both ATP and spermidine into the caspase-cleaved Panx1-containing proteoliposomes (subsection “Pannexin 1 is a conduit for anionic and cationic metabolites”). Thus, flux of one metabolite does not preclude flux of the other and, by comparison with Figure 4B,D, uptake appears to be blunted for both under these conditions (with perhaps a bigger effect on spermidine). We note here that this implies competition of these metabolites for the permeation pathway; it also suggests that the system is not at equilibrium. A kinetic analysis with more time points would be required to determine whether ATP is the preferential permeant in this assay.

8) It would be nice if the authors included a short section in the Discussion considering why other investigators have previously found some permeation through un-cleaved Panx1 channels.

We have now added a short Discussion section noting that some investigators have described permeation through unstimulated Panx1 channels (subsection “Channel properties of Panx1 channels in bilayers and cells”). Unfortunately, as elaborated at length in an earlier review (see Chiu et al., 2018), we do not have a satisfactory explanation for why some groups record such unstimulated channel activity. Thus, we cannot offer much by way of speculation in this regard. We now note that “the mechanisms controlling basal and stimulated channel activity, and the factors that determine the vast range of reported Panx1 channel conductance levels, remain to be determined” (subsection “Channel properties of Panx1 channels in bilayers and cells”).

For our part, we have only observed hPANX1 channel activity after some channel-activating mechanism (i.e., Casp3 cleavage or GPCR signaling) (Billaud et al., 2015; Chiu et al., 2017; Sandilos et al., 2012); the absence of any unstimulated activity also appears to be the case for fPanx1 channels, as demonstrated here. We have seen activity from unstimulated mouse Panx1 channels when they are expressed heterologously (e.g., Sandilos et al., 2012), as others have as well (e.g., Ma et al., 2012; Romanov et al., 2012). Notably, in expression systems those constitutively active mouse channels do not appear to release ATP (Romanov et al., 2012), whereas native mPanx1 channels do not appear to generate basal activity in mouse thymocytes even as caspase-mediated channel activation can provoke ATP release and dye uptake from those same cells (Chekeni et al., 2010; Qu et al., 2011).

Indeed, two important issues in this regard remain unresolved: (1) Do native channels display constitutive activity in the absence of stimulation? and (2) Is any such constitutive activity associated with large molecule permeation?

Recently, Ruan et al., suggested that a side tunnel they identified in the structure of hPANX1 could provide a constitutive atomic ion flux pathway through unstimulated and C-tail intact channels while not allowing large molecule permeation (Ruan et al., 2020). We now mention this possibility in the new discussion of basal channel activity (subsection “Channel properties of Panx1 channels in bilayers and cells”); we had earlier noted this idea when discussing potential separate pathways for atomic ions and large molecules (subsection “Permeant size considerations and pore architecture”).

9) Channel permeation does not depend only on the charge and mass of the permeant but also on its shape (second largest diameter). The authors might include a supplemental figure showing images of the size and shapes of the different permeants that they have tested. It could be briefly addressed in the Discussion.

We have now included a supplemental figure showing the chemical structures of each of the different dyes and metabolites tested in our study, along with their charge and molecular weight (Figure 3—figure supplement 1 and Figure 4—figure supplement 1).

We also now point out that permeation properties of the channel “may be influenced by the shape and flexibility of the permeant molecule and/or the pore itself” (subsection “Permeant size considerations and pore architecture”).

10) Title: How about "ATP and bulky signaling metabolites can flux through Caspase activated Pannexin 1 channels" or something along those lines.

Following this suggestion, we have re-titled the paper as: “ATP and large signaling metabolites flux through caspase-activated Pannexin 1 channels”